# New Evidences of Antibacterial Effects of Cranberry Against Periodontal Pathogens

**DOI:** 10.3390/foods9020246

**Published:** 2020-02-24

**Authors:** María C. Sánchez, Honorato Ribeiro-Vidal, Begoña Bartolomé, Elena Figuero, M. Victoria Moreno-Arribas, Mariano Sanz, David Herrera

**Affiliations:** 1ETEP (Etiology and Therapy of Periodontal Diseases) Research Group, University Complutense of Madrid, Plaza Ramón y Cajal s/n, 28040 Madrid, Spain; mariasb@farm.ucm.es (M.C.S.); hribeiro@ucm.es (H.R.-V.); elfiguer@ucm.es (E.F.); marsan@ucm.es (M.S.); 2Institute of Food Science Research (CIAL), CSIC-UAM, c/ Nicolás Cabrera 9, 28049 Madrid, Spain; b.bartolome@csic.es (B.B.); victoria.moreno@csic.es (M.V.M.-A.)

**Keywords:** polyphenols, cranberry, periodontal diseases, dental biofilm, antibacterial activity, anti-biofilm activity, *F. nucleatum*, *P. gingivalis*, *A. actinomycetemcomitans*

## Abstract

The worrying rise in antibiotic resistances emphasizes the need to seek new approaches for treating and preventing periodontal diseases. The purpose of this study was to evaluate the antibacterial and anti-biofilm activity of cranberry in a validated in vitro biofilm model. After chemical characterization of a selected phenolic-rich cranberry extract, its values for minimum inhibitory concentration and minimum bactericidal concentration were calculated for the six bacteria forming the biofilm (*Streptococcus oralis*, *Actinomyces naeslundii*, *Veillonella parvula*, *Fusobacterium nucleatum*, *Porphyromonas gingivalis*, and *Aggregatibacter actinomycetemcomitans*). Antibacterial activity of the cranberry extract in the formed biofilm was evaluated by assessing the reduction in bacteria viability, using quantitative polymerase chain reaction (qPCR) combined with propidium monoazide (PMA), and by confocal laser scanning microscopy (CLSM), and anti-biofilm activity by studying the inhibition of the incorporation of different bacteria species in biofilms formed in the presence of the cranberry extract, using qPCR and CLSM. In planktonic state, bacteria viability was significantly reduced by cranberry (*p* < 0.05). When growing in biofilms, a significant effect was observed against initial and early colonizers (*S. oralis* (*p* ≤ 0.017), *A. naeslundii* (*p* = 0.006) and *V. parvula* (*p* = 0.010)) after 30 or 60 s of exposure, while no significant effects were detected against periodontal pathogens (*F. nucleatum*, *P. gingivalis* or *A. actinomycetemcomitans* (*p* > 0.05)). Conversely, cranberry significantly (*p* < 0.001 in all cases) interfered with the incorporation of five of the six bacteria species during the development of 6 h-biofilms, including *P. gingivalis*, *A. actinomycetemcomitans*, and *F. nucleatum*. It was concluded that cranberry had a moderate antibacterial effect against periodontal pathogens in biofilms, but relevant anti-biofilm properties, by affecting bacteria adhesion in the first 6 h of development of biofilms.

## 1. Introduction

Dental biofilm-organized periodontal pathogens (including *Porphyromonas gingivalis* and *Aggregatibacter actinomycetemcomitans*) are the primary etiological factor of periodontal diseases, which are one of the most prevalent conditions affecting human beings [1]. These conditions have not only a relevant impact in the mouth [1], but also in systemic health [2] and in quality of life indicators [3]. Due to the infectious nature of periodontal diseases, antimicrobials are widely used in their management (prevention and treatment) [4,5]. However, the worrying rise in antibiotic resistances, including those in periodontal pathogens [6] and unwanted effects of antiseptics/antimicrobials compounds [4,5] emphasize the need to seek new approaches for treating and preventing periodontal diseases. Therefore, attention is given to the need of finding, developing, and improving antimicrobial natural compounds, capable of inhibiting the proliferation and/or adhesion of bacteria pathogens in dental/oral biofilms [5,7,8,9,10]. 

In previous studies, it has been shown that polyphenols, and other compounds derived from plants have an influence on human microbiota, either by promoting the growth of beneficial microorganisms or by acting against pathogens [11,12]. Cranberry (*Vaccinium macrocarpum*) compounds, including phenolic acids, proanthocyanidins (particularly, A-type proanthocyanidins), anthocyanins, organic acids, and their microbial-derived metabolites [13], selectively inhibit the growth of intestinal pathogens such as *Staphylococcus strains* and *Salmonella enterica* [14], reduce *Escherichia coli* colonization of the urinary tract [15,16,17], restrict the virulence of *Pseudomonas aeruginosa* [18,19], present anti-oxidant potential [20], anti-adhesion of Gram-negative and Gram-positive bacteria [21,22], and anti-motility [23,24]. Furthermore, they may be associated with relevant health benefits, including a decreased risk of cardiovascular disease-related mortality [25], prevention of type 2 diabetes mellitus [26], and potential anti-cancer properties [27,28].

The antibacterial and anti-adhesion features of cranberry against oral bacteria have drawn wide attention [22,29,30,31]. Several in vivo and in vitro studies have evaluated how certain cranberry derived compounds could interfere with formation of a cariogenic biofilm. In this regard, it has been demonstrated that certain components of cranberries may limit dental caries by inhibiting the production of organic acids by cariogenic bacteria, the formation of biofilms by *Streptococcus mutans* and *Streptococcus sobrinus*, and the adhesion and coaggregation of a considerable number of other oral species of *Streptococcus* [32,33,34,35,36]. Focusing on periodontal diseases, the non-dialyzable constituent fraction of cranberry (NDM) inhibits the formation of *P. gingivalis* [37] and *Fusobacterium nucleatum* [38] biofilms, two bacteria species associated with periodontitis. The NDM fraction may also inhibit the adhesion of *P. gingivalis* to various proteins, including type I collagen [37] and may reduce bacterial coaggregation involving periodontal pathogens [32]. However, the information on the antibacterial and anti-biofilm capacity of natural extracts from cranberry against relevant periodontal pathogens, growing in complex multi-species biofilms, is scarce. 

Therefore, the aim of the present study was to evaluate the antibacterial and anti-biofilm activity of cranberry extracts in a multispecies in vitro biofilm model, including six bacteria species (*Streptococcus oralis*, *Veillonella parvula*, *Actinomyces naeslundii* and the periodontal pathogens *P. gingivalis*, *A. actinomycetemcomitans*, and *F. nucleatum*). The specific objectives were to assess (1) the antibacterial activity of a cranberry extract against bacteria species in formed biofilms, by assessing the reduction in bacteria viability, and (2) the anti-biofilm activity, by studying the inhibition of the incorporation of different bacteria species in biofilms formed in the presence of the cranberry extract. 

## 2. Materials and Methods 

### 2.1. Cranberry Extract

The cranberry extract used in this study was provided by Triarco Industries Inc. (Cranbury, NJ, USA). For determination of its total polyphenols content, the extract (0.05 g) was dissolved in 10 mL of methanol/HCl (1000:1, *v/v*), sonicated (120 W) for 5 min followed by an extra 15 min resting period, centrifuged, and filtered through a 0.22-μm membrane filter. For analysis of individual phenolic compounds, the extract (0.50 g) was dissolved in 10 mL of MeOH/H_2_O (20:80, *v/v*) containing 0.2% HCl, sonicated for 10 min, centrifuged, and filtered through 0.22 μm. In both cases, sample preparations were performed in duplicate.

### 2.2. Analysis of Phenolic Compounds in the Cranberry Extract

Total polyphenols content was measured by the Folin-Ciocalteu reagent (Merck, Darmstadt, Germany) and using gallic acid (25–500 mg L^−1^) as a calibration standard. Analysis of individual phenolic compounds was carried out by UPLC-DAD-ESI-TQ MS, as previously described in Sanchez-Patán et al. [39]. Different phenolic acids (including phenylpropionic, phenylacetic, mandelic, benzoic, and cinnamic acids), flavan-3-ols (monomers, B-type procyanidin dimers and trimers, and A-type procyanidin dimers and trimers), and anthocyanins (peonidin, cyanidin, and malvidin derivatives) were targeted [39]. Commercial standards of these phenolic acids were used to construct calibration curves for sample quantification [39].

### 2.3. Bacteria Strains and Culture Conditions

Reference strains of *S. oralis* CECT 907T, *V. parvula* NCTC 11810, *A. naeslundii* ATCC 19039, *F. nucleatum* DMSZ 20482, *A. actinomycetemcomitans* DSMZ 8324, and *P. gingivalis* ATCC 33277 were used. These bacteria were grown on blood agar plates (Blood Agar Oxoid No 2; Oxoid, Basingstoke, UK), supplemented with 5% (*v/v*) sterile horse blood (Oxoid), 5.0 mg L^−1^ hemin (Sigma, St. Louis, MO, USA) and 1.0 mg L^−1^ menadione (Merck, Darmstadt, Germany) in anaerobic conditions (10% H_2_, 10% CO_2_, and balance N_2_) at 37 °C for 24–72 h. 

### 2.4. Antibacterial Assays

Figure 1 shows the experimental design followed for the study of the antibacterial effects of cranberry against planktonic bacteria and in an oral biofilm model.

#### 2.4.1. Antibacterial Effect of Cranberry Extract Against Planktonic Bacteria 

Pure cultures of the bacteria species were grown anaerobically in a protein rich medium containing brain-heart infusion (BHI) (Becton, Dickinson and Company, Franklin Lakes, NJ, USA) supplemented with 2.5 g L^−1^ mucin (Oxoid), 1.0 g L^−1^ yeast extract (Oxoid), 0.1 g L^−1^ cysteine (Sigma), 2.0 g L^−1^ sodium bicarbonate (Merck), 5.0 mg L^−1^ hemin (Sigma), 1.0 mg L^−1^ menadione (Merck), and 0.25% (*v/v*) glutamic acid (Sigma). The bacteria growth was harvested at mid-exponential phase (measured by spectrophotometry). Microtitre plate-based antibacterial assays were carried out in a 96-wells plate, adding 190 μL of each bacteria inoculum at a final concentration of 10^6^ colony forming units (CFUs) mL^−1^, and 10 µL of the sterile cranberry extract at a final concentration of 1.0, 0.50, 0.25, 0.10, and 0.01 mg mL^−1^. Plates had a set of controls: negative control (culture media without any inoculum/cranberry extract), positive control (bacteria without any treatment) as well as blanks (cranberry extract or dimethyl sulfoxide (DMSO) dissolved in the culture media), to ensure the validity of the assay, 4% DMSO (to identify a possible bactericidal effect of DMSO, used as a solvent for the cranberry extract), and 0.2% chlorhexidine (CHX), in order to compare with the reference of known antibacterial effect. A measurement (optical density, O.D._595_) as t = 0 absorbance was taken in a microtitre plate reader (Optic Ivymen System 2100-C; I.C.T.; La Rioja, Spain). The microplates were incubated for 48 h at 37 °C under anaerobic conditions, and absorbance was measured at selected intervals (1 h during the first 12 h, and every 12 h to complete 48 h), in order to determine the bacteria growth in time, until the bacteria reached stationary growth phase. MIC (minimum inhibitory concentration) and MBC (minimum bactericidal concentration) values were calculated and confirmed by microbial plate counting on blood agar media. Accordingly, the lowest concentration of the cranberry extract showing growth inhibition was considered as the MIC, whereas the lowest concentration of the cranberry extract that showed zero growth in blood agar plates, after spot inoculation and incubation for 72 h, was recorded as the MBC. All experiments were performed in triplicate with appropriate controls. 

#### 2.4.2. Antibacterial Effect in an Oral Biofilm Model in Vitro

In order to optimize the method for evaluating the antibacterial effect of the cranberry extract against the bacteria species growing in biofilms, a range of cranberry concentrations were initially tested (from MBCs to stock solution of cranberry extracts at 20 mg mL^−1^). A dose of 20 mg mL^−1^ yielded the higher antibacterial effect (data not shown). 

The multi-species in vitro biofilm model was developed as previously described by Sánchez et al. [40]. Briefly, pure cultures of each bacteria specie were grown anaerobically in the supplemented BHI medium. The bacteria growth was harvested at mid-exponential phase (measured by spectrophotometry), and a mixed bacteria suspension in modified BHI medium containing 10^3^ CFUs mL^−1^ for *S. oralis*, 10^5^ CFUs mL^−1^ for *V. parvula* and *A. naeslundii*, and 10^6^ CFUs mL^−1^ for *F. nucleatum*, *A. actinomycetemcomitans* and *P. gingivalis* was prepared (different concentrations based on the different growth rates of each bacteria species). Sterile calcium hydroxyapatite (HA) discs of 7 mm of diameter and 1.8 mm (standard deviation, SD = 0.2) of thickness (Clarkson Chromatography Products, Williamsport, PA, USA) were coated with sterile saliva for 4 h at 37 °C in sterile plastic tubes to allow the formation of the acquired pellicle [40], and then placed in the wells of a 24-well tissue culture plate (Greiner Bio-one, Frickenhausen, Germany). Each well was inoculated with 1.5 mL mixed bacteria suspension prepared and incubated in anaerobic conditions (10% H_2_, 10% CO_2_, and balance N_2_) at 37 °C for 72 h. 

After 72 h, biofilms were dipped during 30 s and 60 s in the cranberry solution (20 mg mL^−1^), at room temperature. Exposure time of 30 and 60 s were selected since cranberry extracts are bioactive products, commercially available, and for them, the standard exposure times established for other antimicrobial commercially available products (e.g., chlorhexidine mouth rinses), were selected [41,42,43]. Phosphate buffered saline solution (PBS) was used as negative control and, in order to discard a bactericidal effect of DMSO used to dissolve the extracts, a 4% DMSO solution was also tested.

The antibacterial activity in 72 h biofilms was examined by determining the reduction in the number of viable bacteria counts (expressed as CFUs mL^−1^), using quantitative polymerase chain reaction (qPCR) combined with Propidium Monoazide (PMA), and by Confocal Laser Scanning Microscopy (CLSM). Assays were conducted in triplicate (with trios of biofilms per replica).

### 2.5. Anti-Biofilm Assay

In order to optimize the method for evaluating the anti-biofilm effect of cranberry extracts against the selected bacteria species, different concentrations were tested, based on MICs of each bacteria species in planktonic state (data not shown), and it was finally concluded that a dose of 0.20 mg mL^−1^ provided the largest anti-biofilm impact, without affecting bacteria viability in planktonic state.

For the anti-biofilm assay, the mixed bacteria suspension in modified BHI medium containing 10^3^ CFUs mL^−1^ for *S. oralis*, 10^5^ CFUs mL^−1^ for *V. parvula* and *A. naeslundii*, and 10^6^ CFUs mL^−1^ for *F. nucleatum*, *A. actinomycetemcomitans* and *P. gingivalis* was prepared as previously described. HA discs were coated with treated saliva for 4 h at 37 °C in sterile plastic tubes, and then placed in the wells of a 24-well tissue culture plates. Each well was inoculated with 1.5 mL mixed bacteria suspension prepared and the cranberry extract at 0.20 mg mL^−1^, or with PBS and DMSO in control biofilms, were added. Plates were incubated in anaerobic conditions, at 37 °C for 6 h. 

The anti-biofilm activity was examined by determining bacteria counts in biofilms, as CFUs mL^−1^ by means of qPCR, and by CLSM. Assays were conducted in triplicate (with trios of biofilms per replica).

### 2.6. Microbiological Outcomes

After antibacterial and anti-biofilm assays, biofilms were recovered and sequentially rinsed in 2 mL of sterile PBS (immersion time per rinse, 10 s) three times, in order to remove possible remnants of the extracts and non-adherent bacteria. Then, biofilms were disrupted by vortex for 2 min in 1 mL of PBS. In the case of antibacterial activity, and to discriminate between DNA from live and dead bacteria, PMA was used (Biotium Inc., Hayword, CA, USA). The use of this PMA dye has shown the ability to distinguish between viable and irreversibly damaged cells and hence when combined with qPCR to detect the DNA from viable bacteria [44]. PMA was added to sample tubes containing 250 µL of disaggregated biofilm cells, at a final concentration of 100 µM. Following an incubation period of 10 min at 4 °C in the dark, the samples were subjected to light-exposure for 30 min, using PMA-Lite LED Photolysis Device (Biotium Inc.). After PMA photo-induced DNA cross-linking, the cells were centrifuged at 15,000 rcf for 3 min prior to DNA isolation. 

Bacteria DNA was isolated from all biofilms using a commercial kit ATP Genomic DNA Mini Kit^®^ (ATP biotech. Taipei, Taiwan), following manufacturer’s instructions and the hydrolysis 5’nuclease probe assay qPCR method was used for detecting and quantifying the bacteria DNA. The qPCR amplification was performed following a protocol previously optimized by our research group, using primers and probes targeted against 16S rRNA gene (obtained through Life Technologies Invitrogen (Carlsbad, CA, USA) and Applied Biosystems (Carlsbad, CA, USA)) [44]. 

Each DNA sample was analyzed in duplicate. Quantification cycle (Cq) values, previously known as cycle threshold (Ct) values, describing the PCR cycle number at which fluorescence rises above the baseline, were determined using the provided software package (LC 480 Software 1.5; Roche Diagnostic GmbH; Mannheim, Germany). Quantification of viable cells by qPCR was based on standard curves. The correlation between Cq values and CFUs mL^−1^ was automatically generated through the software (LC 480 Software 1.5; Roche).

All assays were developed with a linear quantitative detection range established by the slope range of 3.3–3.7 cycles/log decade, r^2^ > 0.998, and an efficiency range of 1.9–2.0.

### 2.7. Confocal Laser Scanning Microscopy (CLSM) Analyses 

After antibacterial and anti-biofilm treatment referred above, and before CLSM analysis, the discs were sequentially rinsed in 2 mL of sterile PBS (immersion time per rinse, 10 sec) three times, in order to remove possible remnants of the extract and non-adherent bacteria. Non-invasive confocal imaging of fully hydrated biofilms was carried out using a fixed-stage Ix83 Olympus inverted microscope coupled to an Olympus FV1200 confocal system (Olympus; Shinjuku, Tokyo, Japan). Specimens were stained with LIVE/DEAD^®^ BacLight^TM^ Bacteria Viability Kit solution (Molecular Probes B. V., Leiden, The Netherlands) at room temperature. The 1:1 fluorocrome ratio with a staining time of 9 ± 1 min was used to obtain the optimum fluorescence signal at the corresponding wave lengths (Syto9: 515–530 nm; Propidium Iodide (PI): >600 nm). At least three separate and representative locations on the discs covered with biofilm were selected for these measurements (based on the presence of stacks or “towers” identified in the confocal field view). The CLSM software was set to take a z-series of scans (xyz) of 1 µm thickness (8 bits, 1024 × 1024 pixels). Image stacks were analyzed by using the Olympus^®^ software (Olympus). Image analysis and live/dead cell ratio (i.e., the area occupied by living cells divided by the area occupied by dead cells) was performed with Fiji software (ImageJ Version 2.0.0-rc-65/1.52b, Open source image processing software).

### 2.8. Statistical Analyses

The selected outcome variables to study the antibacterial effect of cranberry extracts were the counts of viable bacteria present on the biofilms, expressed as viable CFUs mL^−1^ of *S. oralis*, *V. parvula*, *A. naeslundii*, *F. nucleatum*, *A. actinomycetemcomitans*, and *P. gingivalis* by qPCR, and the live/dead cell ratio of the whole biofilm by CLSM. An experiment-level analysis was performed for each parameter of the study (n = 9 for qPCR and n = 3 for CLSM results). Shapiro–Wilk goodness-of-fit tests and distribution of data were used to assess normality. The effect of each solution (cranberry extracts, PBS and 4% DMSO), the time of exposure (30 or 60 s), and their interaction with the main outcome variable (counts expressed as CFUs mL^−1^ or live/dead cell ratio), was compared by means of a parametric ANOVA test for independent samples, and a general linear model was constructed for each bacterium for qPCR results and for total bacteria for live/dead cell ratio of whole biofilm obtained by CLSM, using the method of maximum likelihood and Bonferroni corrections for multiple comparisons.

To study the anti-biofilm effect of the cranberry extract, the selected outcome variables were the counts of bacteria present on the biofilms, expressed as CFUs mL^−1^ of *S. oralis*, *V. parvula*, *A. naeslundii*, *F. nucleatum*, *A. actinomycetemcomitans*, and *P. gingivalis* by qPCR, and the live/dead cell ratio of the whole biofilm by CLSM. Shapiro–Wilk goodness-of-fit tests and distribution of data were used to assess normality. An experiment-level analysis was performed for each parameter of the study (n = 9 for qPCR and n = 3 for CLSM results). The effect of each solution (cranberry extract, PBS, and 4% DMSO) on the main outcome variables (CFUs mL^−1^ or live/dead cell ratio), was compared by means of a parametric ANOVA test for independent samples, using the method of maximum likelihood and Bonferroni corrections for multiple comparisons.

Data was expressed as means ± SD and as the mean percent inhibition that was calculated by Equation: Percent inhibition = (CFUs mL^−1^ of negative control–CFUs mL^−1^ of test / CFUs mL^−1^ of negative control) × 100.

Results were considered statistically significant at *p* < 0.05. A software package (IBM SPSS Statistics 24.0; IBM Corporation, Armonk, NY, USA) was used for all data analysis. 

## 3. Results

### 3.1. Phenolic Composition of the Cranberry Extract

A phenolic characterization of the cranberry extract was initially carried out to ensure its susceptibility for this study. The content in total polyphenols content of the extract resulted in 219 mg of gallic acid equivalents g^−^^1^. Concerning phenolic composition, Table 1 details the main phenolic compounds found in the extract, as determined by UPLC-DAD-ESI-TQ MS. Among phenolic acids (benzoic and cinnamic acids), the extract was especially rich in benzoic acid (8.38 mg g^−^^1^), followed by others such as p-coumaric acid (0.84 mg g^−1^) and protocatechuic acid (0.73 mg g^−1^) in considerable less content. Concerning flavan-3-ols, main MS signals corresponded to A-type trimers (1.58 mg g^−1^), followed by A-type (0.23 mg g^−1^) and B-type (0.20 mg g^−1^) dimers, monomers (0.065 mg g^−1^), and B-type trimers (0.034 mg g^−1^). In relation to anthocyanins, peonidin-3-arabinoside (0.32 mg g^−1^) and cyanidin-3-arabinoside (0.15 mg g^−1^) showed the highest content (Table 1). These compositional data were in accordance to others commercial cranberry extracts [39].

### 3.2. Antibacterial Assays

#### 3.2.1. Antibacterial Effect of Cranberry Extract Against Planktonic Bacteria

MICs and MBCs values against the six bacteria species selected in planktonic state were determined for the selected cranberry extract. MICs indicated an average bacteriostatic concentration of 0.10 mg mL^−1^ against *P. gingivalis* and *F. nucleatum*, 0.25 mg mL^−1^ for *A. naeslundii* and *A. actinomycetemcomitans*, 0.50 mg mL^−1^ for *V. parvula*, and >1.00 mg mL^−1^ for *S. oralis*. MBCs tests showed similar results, with bactericidal concentrations of 0.25 mg mL^−1^ against *P. gingivalis*, 1.00 mg mL^−1^ against *F. nucleatum*, and >1.00 mg mL^−1^ for *S. oralis, A. naeslundii, V. parvula,* and *A. actinomycetemcomitans*. According to these results, the cranberry extract exhibited antibacterial activity, displaying the largest antibacterial properties against the periodontal pathogens *P. gingivalis* and *F. nucleatum.*

#### 3.2.2. Antibacterial Effects in an in Vitro Biofilm Model: Bacteria Counts

Table 2 depicts the effect of cranberry extracts (20 mg mL^−1^), compared to the negative control (PBS) and 4% DMSO control solution, on the mean number of viable bacteria counts in 72 h biofilms. After an exposure of 30 or 60 s to the cranberry extract, significant reductions in viable counts in biofilms were observed for initial and early colonizers. *S. oralis* showed significant reductions after 30 (*p* < 0.001) and 60 s (*p* = 0.017) when compared to negative control (PBS), reaching in both cases a decrease of 98.9% of viable CFUs (Table 2). Significant differences (*p* < 0.001 after 30 s of exposure) were also observed when the effects of DMSO solution was compared to PBS, with percentages of decrease of 93.1% and 58.8% for 30 and 60 s exposures, respectively (Table 2). For *A. naeslundii* and *V. parvula*, a significant impact of the cranberry extract was observed after 30 s (65.7% of reduction, *p* = 0.006 and 66.7% of reduction, *p* = 0.010, respectively), but not after 60 s. No significant reductions were observed after exposure to DMSO (*p* > 0.05), after 30 or 60 s (Table 2). No statistically significant differences were observed between the cranberry extract and DMSO at any time (*p* > 0.05). The effect of exposure time (30 s versus 60 s) was not statistically significant for both solutions (*p* > 0.05 in all cases) in *S. oralis, A. naeslundii* and *V. parvula*.

For the secondary colonizer *F. nucleatum*, some effects on viable counts were observed after 30 s (*p* = 0.164) and after 60 s (decrease of 75.3%, *p* = 0.448), although not statistically significant. Additionally, no statistically significant reductions in viable counts were observed for DMSO (*p* > 0.05) (Table 2). No statistically significant differences were observed between the cranberry extract and DMSO at any time (*p* > 0.05). The effect of exposure time was however, statistically significant for the cranberry extract (*p* = 0.022) and DMSO (*p* = 0.035).

For the periodontal pathogens *A. actinomycetemcomitans* and *P. gingivalis*, no significant reductions in viable counts after 30 s or 60 s of exposure to the cranberry extract (*p* > 0.05) were observed when compared to negative control: reductions of 11.5% for *A. actinomycetemcomitans* and 39.3% for *P. gingivalis* after 60 s. The same was true for DMSO (*p* > 0.05) (Table 2). No statistically significant differences were observed in the effectiveness comparing the two solutions at applied times or when comparing exposure times (*p* > 0.05 for all cases).

#### 3.2.3. Antibacterial Effects in an in Vitro Biofilm Model: CLSM

The CLSM analysis showed that, after 72 h of incubation on HA surfaces, control biofilms covered the entire disc surface as a flat layer of cells combined with stacks of bacteria aggregations, showed a live/dead cell ratio (i.e., the area occupied by living cells divided by the area occupied by dead cells) of 1.43 (SD 0.10) and 1.25 (SD 0.15), after exposure of 30 and 60 s, respectively, to PBS (Figure 2a, b). Table 3 depicts the effects of the cranberry extract on the live/dead cell ratio of the whole biofilm obtained by CLSM. It could be observed that, after exposure of 30 s to cranberry extracts and to the 4% DMSO solution, cell vitality significantly decreased in the biofilms, showing, respectively, live/dead cell ratios of 0.67 (SD 0.07) and 0.77 (SD 0.04) for 4% DMSO (*p* < 0.001 in both cases, when compared to negative control biofilms) (Figure 2c, e). After 60 s of exposure (Figure 2f), reductions in viability were also statistically significant for cranberry extracts (live/dead cell ratio of 0.56 (SD 0.02), *p* < 0.001; Figure 2f) and for DMSO solution (live/dead cell ratio of 0.78 (SD 0.05), *p* < 0.001; Figure 2d), when compared to control biofilms (live/dead cell ratio of 1.25 (SD 0.15)). These results are consistent with those observed by means of qPCR, with significant differences in viable counts of initial and early colonizers, after exposure to cranberry extracts and DMSO solution, when compared to negative control biofilms. Statistically significant differences were observed between the cranberry extract and DMSO after 60 s of exposure (*p* = 0.027) (Table 3).

### 3.3. Anti-Biofilm Assay

#### 3.3.1. Anti-Biofilm Assay: Bacteria Counts

The cranberry extract, at a concentration of 0.20 mg mL^−1^, significantly inhibited the incorporation of five of the six studied bacteria species in the in vitro biofilm model (Table 4). After 6 h of contact, and compared to negative control biofilms, two of the three initial and early colonizers were significantly reduced on the HA surfaces: 98.9% for *S. oralis* (*p* < 0.001) or 90.9% for *V. parvula* (*p* < 0.001), when exposed to cranberry extracts. No significant impact was observed for *A. naeslundii*.

Periodontal pathogens showed a similar trend. *P. gingivalis* showed the largest impact of cranberry extracts: 97.2% (*p* < 0.001), with counts of 1.1 × 10^3^ (SD 1.1 × 10^3^) CFUs mL^−1^, compared to 4.0 × 10^4^ (SD 2.9 × 10^4^) CFUs mL^−1^, in negative control biofilms. Reductions *A. actinomycetemcomitans* (84.0%) and *F. nucleatum* (75.4%) were statistically significant (*p* < 0.001 in both cases).

For DMSO, a significant impact was observed for the three periodontal pathogens and for *S. oralis*, when compared to control biofilms (*p* < 0.005 in all cases; Table 4).

Significant differences were observed in the effectiveness comparing the cranberry extract and DMSO solution after 6 h of biofilm evolution in *V. parvula* (*p* < 0.001) and *A. actinomycetemcomitans* (*p* = 0.024).

#### 3.3.2. Anti-Biofilm Assay: CLSM

CLSM analysis showed that, after 6 h of incubation on HA surfaces, formed biofilms showed the typical features of bacteria communities in their first steps, with a high percentage of live cells versus dead cells, that was evidenced by a live/dead cell ratio of 1.44 (SD 0.01) (Figure 3a,b). The effect of the exposure of the biofilms for 6 h to the cranberry extract, at a concentration of 0.20 mg mL^−1^, was evident as it was not possible to observe well-structured biofilms, contrary what happened in the control samples. Although the biomass was reduced, no significant differences in bacteria vitality were observed when compared respect to controls (live/dead cell ratio of 0.99 (SD 0.01), *p* = 0.160; Table 3; Figure 3c,d), suggesting a limited antiseptic effect, and highlighting a desired effect on bacteria adhesion. Conversely, DMSO showed a similar live/dead cell ratio (1.047 (SD 0.14); Figure 3e,f), and biofilms were normally formed, with no significant differences when compared to control biofilms (*p* = 0.101; Table 3). These results are consistent with those observed by qPCR, which showed significant differences in bacteria counts of the tested bacteria species when biofilms formed in the presence of the cranberry extract where compared with control biofilms.

No statistically significant differences were observed in the effectiveness comparing the cranberry extract and DMSO at applied time (*p* = 1.000) (Table 3).

## 4. Discussion

Since bacteria resistance to antibiotics is becoming an increasing health threat worldwide, alternative strategies to prevent or limit biofilm formation are a relevant goal. A growing body of evidence has demonstrated that plant extracts offer relevant antimicrobial and anti-biofilm potentials, with no significant risk of increasing antibiotic resistances. A vast number of phytochemicals have been recognized as valuable alternatives and complementary products to manage bacterial infections [45,46]. Cranberry (*Vaccinium macrocarpum*) fruits are particularly rich in biologically active phenolic compounds and organic acids [13], as it has also been confirmed from our results (Table 1). Numerous in vivo and in vitro studies have showed that different cranberry compounds/fractions/extracts possess antibacterial properties (against both Gram-positive and Gram-negative bacteria species) on various pathogenic bacteria in urinary tract infections and other diseases [33,47,48,49]. In this context, the present study has confirmed the antibacterial capacity of cranberry extracts against the six bacteria species (*S. oralis* CECT 907T, *V. parvula* NCTC 11810, *A. naeslundii* ATCC 19039, *F. nucleatum* DMSZ 20482, *A. actinomycetemcomitans* DSMZ 8324, and *P. gingivalis* ATCC 33277) tested in planktonic state; this findings contradict, at least partially, those of La and co-workers [50], who concluded that A-type proanthocyanidins did not present any effect on *P. gingivalis* in planktonic state. In view of our results from the UPLC-DAD-ESI-TQ MS analysis of the cranberry extract, a possible explanation for this disagreement could be that the antiseptic effect comes, not only from these A-type proanthocyanidins, but also from some of the other components of cranberry extracts, such as phenolic acids. In this context, the necessity of carrying out previous compositional characterization of cranberry extracts should be noted—as we have done in our study—in view of the diversity in this kind of products that affects their bioactivity [51].

However, bacteria are normally arrange as biofilms, predominating these sessile communities in most of the environmental, industrial, and medical habitats [52]. In fact, these highly structured bacteria communities are found in the mouth, allowing bacteria cells to withstand the natural defence mechanisms, as well as the host’s immune defences or the effects of antimicrobial agents [53,54,55]. Therefore, the study of the antimicrobial of cranberry extracts should be performed against bacteria organized in biofilms. The results of the present study indicate that, when testing bacteria organized in biofilms, bacteria viability was affected by exposure to the cranberry extract at 20 mg mL^−1^ after 30 and 60 s of exposure. However, a significant effect was only observed for initial and early colonizers (*S. oralis, A. naeslundii*, and *V. parvula*), but, in agreement with other studies, not for periodontal pathogens (*F. nucleatum, P. gingivalis*, and *A. actinomycetemcomitans*). Philips and coworkers [56], in a recent investigation assessing the inhibitory effects of berry fruit extracts on *S. mutans* biofilms, indicated that bacteria viability was not significantly affected, as also concluded by Koo and coworkers [33]. Biofilms are an intriguing structure which demonstrate greater resistance to antimicrobial agents when compared to organisms in planktonic form [31]. A previous study using a cranberry juice concentrate formulated as a thermoreversible gel [11], showed antibacterial properties against *A. actinomycetemcomitans* and *P. gingivalis*, in contrast to the results of our study. The variability of the results may be due to the different types of samples and formulations used.

Besides the antibacterial effects, this investigation highlights new possible features regarding the anti-biofilm activity of cranberry extracts against periodontal pathogens. Bacteria adhesion to oral surfaces is the initial and crucial step in dental biofilm development and, therefore, in the pathogenesis of periodontal diseases. The cranberry extract, at a concentration of 0.20 mg mL^−1^, inhibited the colonization of the six tested bacteria species in the in vitro biofilm model, especially for periodontal pathogens *P. gingivalis* (97.2% of reduction), *A. actinomycetemcomitans* (84%), and *F. nucleatum* (75.4%), being the impact statistically significant (*p* < 0.001 in all cases), when compared to control biofilms. Additionally, initial and early colonizers were significantly affected: *S. oralis* (98.9%, *p* < 0.001) or *V. parvula* (90.9%, *p* < 0.001). Different studies have described the role of cranberry constituents in bacteria adhesion and biofilm development: Philips and coworkers [56] indicated that cranberry extracts were the most effective extract in disrupting *S. mutans* biofilm integrity and structural architecture, without significantly affecting bacteria viability; La and co-workers [50] observed that A-type cranberry proanthocyanidins did not have any effect on *P. gingivalis* planktonic growth, but they did inhibit biofilm formation. The anti-biofilm effect of cranberry extracts in our biofilm model was also confirmed by CLSM, with a significant disturbance on biofilm structure, a qualitative assessment that was consistent with the quantitative data provided by qPCR.

Labreque and coworkers [37] and Yamanaka and coworkers [38] observed that the non-dialyzable constituent fraction of cranberry (NDM) interfered with the colonization of *P. gingivalis* and *F. nucleatum* in the gingival crevice, reducing bacteria coaggregation in periodontal diseases [37,38,57,58]. Moreover, Polak et al. [58] found that NDM adhesion of *P. gingivalis* and *F. nucleatum* onto epithelial cells, and NDM consumption by mice attenuated the severity of experimental periodontitis, compared with a mixed infection without NDM treatment. Furthermore, NDM increased the phagocytosis of *P. gingivalis*. In addition, cranberries were described to restrain the proteolytic activity of the red complex, specifically the gingipain activity of *P. gingivalis*, trypsin-like activity of *Tannerella forsythia*, and chemotrypsin-like activity of *Treponema denticola* [59]. Cranberry extracts have also demonstrated the inhibition of the productions some cytokines: Bodet et al. [59] or Polak et al. [58] observed that NDM eliminated TNF-a expression by macrophages that were exposed to *P. gingivalis* and *F. nucleatum*, without impairing their viability. 

The hydrophobic character of the cranberry extract has made the experiments difficult, requiring the use of the organic solvent DMSO in the tests, in order to overcome such complications. However, some antimicrobial activity of DMSO at the selected concentration (4%) was observed, and therefore, it may have a possible contribution in the antibacterial activity of the extract under investigation. In this way, studies have tested different concentrations, ranging up to 10% [60,61,62,63]. However, when used as a solvent, there is no established criteria as to which is the most appropriate concentration, and the interpretation of its effects on the microorganisms with which it interacts is of great importance in view of its widespread use as solvent in therapeutic and pharmacological studies [60,61,62,63]. In the present study, the 4% DMSO concentration was selected as the one that ensured complete solubilisation of the cranberry extract with minimum antimicrobial effects. However, in any case, the results obtained in the present study make evident the need to standardize an appropriate concentration of DMSO, suitable for bacterial experiments, considering that there is a discrepancy in the findings of different studies on the antimicrobial effects of different concentrations of DMSO.

## 5. Conclusions

This study has demonstrated that the incorporation of bacteria into the biofilm was significantly interfered, including relevant periodontal pathogens, such as *P. gingivalis, A. actinomycetemcomitans* and *F. nucleatum*. Our results support the hypothesis that cranberry components may interfere in the phase of bacteria adherence, disabling or inhibiting the adherence of periodontal pathogens and, therefore, preventing bacterial colonization. This fact could interfere with biofilm formation and possibly helping to maintain homeostasis and, thus, to prevent periodontal diseases. Anti-biofilm activity of cranberry extracts in the present study could be attributed to the presence of polyphenols, specifically phenolic acids and A-type proanthocyanidins, which are known to inactivate glucosyl-transferase and fructosyl-transferase that catalyse the formation of glucan and fructan, respectively, which play prime roles in biofilm formation and maturation [31]. It has also been reported that the polyphenols in cranberries led to desorption of biofilm by interfering with bacteria coaggregation [64]. Moreover, cranberries are supposed to reduce periodontal-related symptoms by suppressing inflammatory cascades as an immunologic response to bacteria invasion. 

Despite the limitations of the study, and the great effect caused by the DMSO solvent, the research performed has identified an important anti-biofilm effect of cranberry on periodontal bacteria and serve as a support for the development of further studies, assessing the most effective vehicle and the ideal concentration to be used, without causing adverse effects on oral tissues.

## Figures and Tables

**Figure 1 foods-09-00246-f001:**
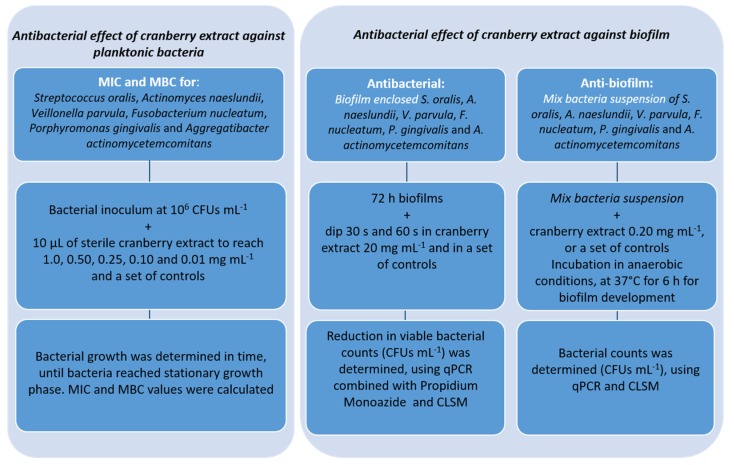
Scheme of the antibacterial assays carried out in this study.

**Figure 2 foods-09-00246-f002:**
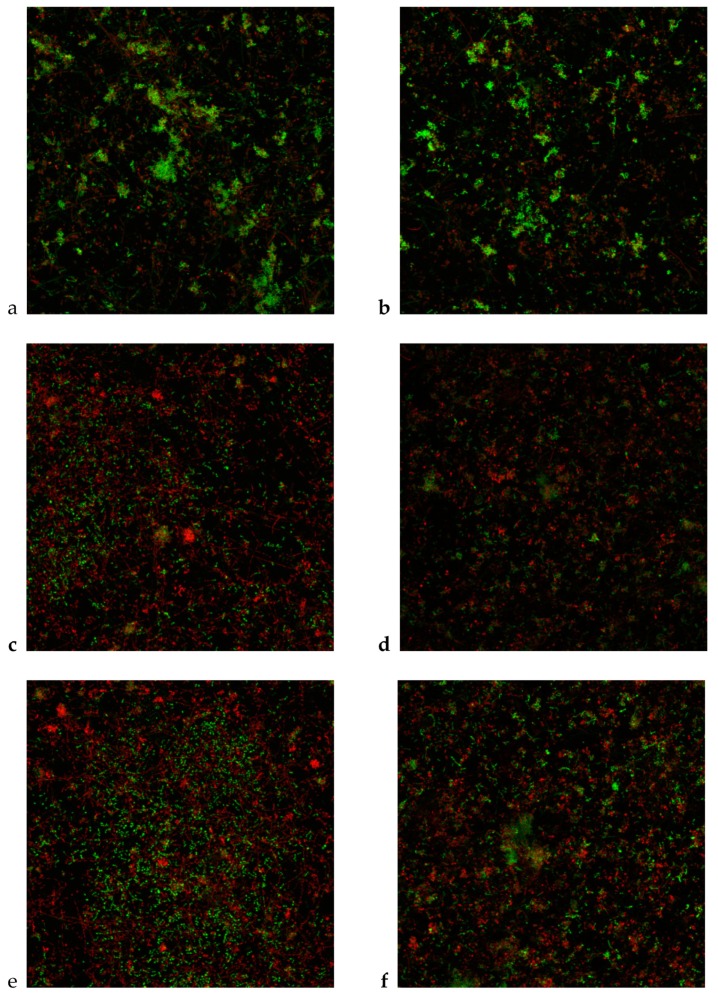
Maximum projection of confocal laser scanning microscopy (CLSM) images of the whole biofilm, grown 72 h over hydroxyapatite surfaces, and stained with LIVE/DEAD^®^ BacLight^TM^ Bacteria Viability Kit, after exposure to: (**a**,**b**) negative controls, 30 and 60 s, respectively (phosphate buffer saline, PBS); (**c**,**d**) 4% dimethyl sulfoxide (DMSO) solution, 30 and 60 s, respectively; (**e**,**f**) cranberry extracts (20 g L^−1^), 30 and 60 s, respectively. (Scale bar = 100 µm).

**Figure 3 foods-09-00246-f003:**
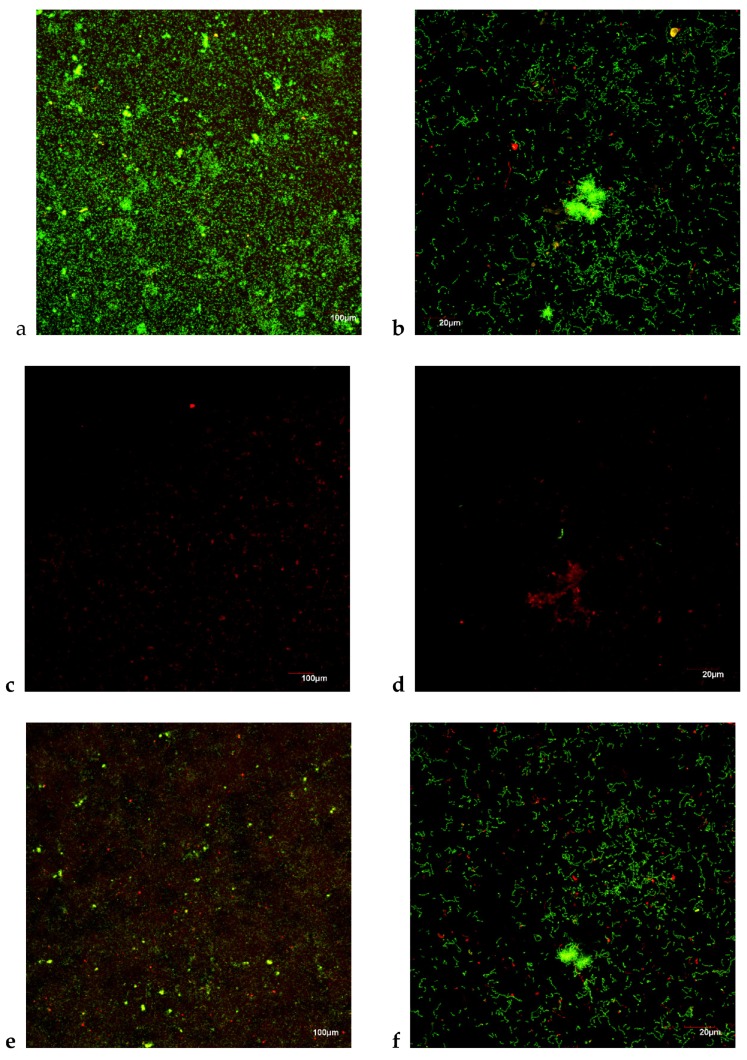
Maximum projection of confocal laser scanning microscopy (CLSM) images of the whole biofilm after 6 h of development, growing in the presence of 0.20 mg mL^−1^ of cranberry extract, over hydroxyapatite surfaces, and stained with LIVE/DEAD^®^ BacLight^TM^ Bacteria Viability Kit, after exposure to: (**a**,**b**) negative control (phosphate buffer saline, PBS); (**c**,**d**) cranberry extract; (**e**,**f**) 4% dimethyl sulfoxide (DMSO) solution.

**Table 1 foods-09-00246-t001:** Phenolic compounds present in the cranberry extract used in this study. Data are expressed as mean and standard deviation (SD).

Compounds Group	Phenolic Compound	Concentration (μg g^−1^ ± SD)
**Benzoic acids**	Benzoic acid	8317.88 ± 222.31
	Protocatechuic acid	735.12 ± 17.76
	Vanillic acid	262.54 ± 10.16
	Gallic acid	136.16 ± 1.50
	4-Hydroxybenzoic acid	94.81 ± 2.23
	Salycilic acid	91.05 ± 2.16
	4-Hydroxymandelic acid	30.84 ± 1.14
	3-O-methylgallic acid	30.05 ± 0.64
	4-Hydroxy-3-methoxymandelic acid	14.33± 0.45
	Syringic acid	11.80 ± 1.18
	3-Hydroxybenzoic acid	11.58 ± 0.01
	3-(3,4-Dihydroxyphenyl)-propionic acid	9.61 ± 0.16
	4-Hydroxy-3-methoxyphenylacetic acid	6.55 ± 0.66
	3,4-Dihydroxy mandelic acid	3.56 ± 0.31
	4-Hydroxypheny lacetic acid	3.20 ± 0.41
	Hippuric acid	1.14 ± 0.10
	3,4-Dihydroxy phenylacetic acid	1.05 ± 0.10
	3,4,5-Trimethoxy benzoic acid	0.32 ± 0.03
**Cinnamic acids**	*p*-Coumaric acid	844.16 ± 15.20
	*trans*-Cinnamic acid	260.55 ± 0.04
	Caffeic acid	133.67 ± 2.52
	Ferulic acid	111.92 ± 4.38
	Trimethoxycinnamic acid	2.72 ± 0.27
**Flavan-3-ols**	∑ A-type trimers	1579.04 ± 27.31
	∑ A-type dimers	230.95 ± 18.11
	∑ B-type dimers	201.87 ± 17.21
	∑ Monomers	65.81 ± 5.20
	∑ B-type trimers	34.1 ± 0.91
**Anthocyanins**	Peonidin-3-arabinoside	32.73 ± 3.27
	Cyanidin-3-arabinoside	15.01 ± 0.05
	Peonidin-3-glucoside	4.84 ± 0.48
	Malvidin-3-arabinoside	1.16 ± 0.02
	Peonidin-3-galactoside	1.03 ± 0.09
	Cyanidin-3-glucoside	0.31 ± 0.02
	Cyanidin-3-galactoside	0.19 ± 0.01

**Table 2 foods-09-00246-t002:** Antibacterial effects of the cranberry extract on the mean number of viable bacteria counts in the in vitro multi-species biofilm model (in colony forming units, CFUs mL^−1^, determined by quantitative polymerase chain reaction (qPCR)). Data are expressed as mean and standard deviation (SD). PBS: phosphate buffer saline; DMSO: 4% dimethyl sulfoxide solution.

	Exposure Time (seconds)	Viable CFUs mL^−1^ [mean (SD)]			*p*-Value When Compared to Negative Control		% of Reduction of viable CFUs mL^−1^ as Compared with Negative Control	
		Negative Control (PBS)	Cranberry Extract	DMSO	Cranberry Extract	DMSO	Cranberry Extract	DMSO
***S. oralis***	30	1.2 × 10^6^ ± 1.1 × 10^6^	1.3 × 10^4^ ± 1.1 × 10^4^	8.3 × 10^4^ ± 1.4 × 10^5^	0.000	0.000	98.9	93.1
	60	6.8 × 10^5^ ± 4.3 × 10^5^	7.3 × 10^3^ ± 4.4 × 10^3^	2.8 × 10^5^ ± 2.3 × 10^5^	0.017	0.282	98.9	58.8
***A. naeslundii***	30	6.7 × 10^4^ ±5.6 × 10^4^	2.3 × 10^4^ ± 1.3 × 10^4^	3.4 × 10^4^ ± 2.1 × 10^4^	0.006	0.050	65.7	49.2
	60	2.2 × 10^4^ ±1.3 × 10^4^	3.2 × 10^4^ ± 2.4 × 10^4^	2.0 × 10^4^ ± 1.4 × 10^4^	1.000	1.000	45.4	9.1
***V. parvula***	30	3.6 × 10^6^ ±2.8 × 10^6^	1.2 × 10^6^ ± 1.4 × 10^6^	2.0 × 10^6^ ± 2.1 × 10^6^	0.010	0.147	66.7	44.4
	60	1.6 × 10^6^ ± 8.6 × 10^5^	4.4 × 10^5^ ± 3.6 × 10^5^	1.3 × 10^6^ ± 1.3 × 10^6^	0.395	1.000	72.5	18.7
***A. actinomycetemcomitans***	30	7.2 × 10^6^ ± 6.4 × 10^6^	6.8 × 10^6^ ± 4.7 × 10^6^	5.6 × 10^6^ ± 3.0 × 10^6^	1.000	1.000	5.6	22.2
	60	5.2 × 10^6^ ± 3.5 × 10^6^	4.6 × 10^6^ ± 4.4 × 10^6^	5.2 × 10^6^ ± 4.9 × 10^6^	1.000	1.000	11.5	0.0
***P. gingivalis***	30	1.7 × 10^6^ ± 7.0 × 10^5^	1.1 × 10^6^ ± 5.2 × 10^5^	1.6 × 10^6^ ± 1.8 × 10^6^	0.434	1.000	35.3	5.9
	60	8.9 × 10^5^ ± 6.8 × 10^5^	5.4 × 10^5^ ± 1.8 × 10^5^	1.0 × 10^6^ ± 7.0 × 10^5^	1.000	1.000	39.3	12.3
***F. nucleatum***	30	3.8 × 10^5^ ± 3.1 × 10^5^	2.3 × 10^5^ ± 1.5 × 10^5 †^	3.5 × 10^5^ ± 1.3 × 10^5^	0.164	1.000	39.5	7.9
	60	1.5 × 10^5^ ± 1.0 × 10^5^	3.7 × 10^4^ ± 3.0 × 10^4 †^	1.8 × 10^5^ ± 1.5 × 10^5^	0.448	1.000	75.3	0.0

^†^*p* < 0.05, significant differences when comparing exposure times for an antimicrobial agent.

**Table 3 foods-09-00246-t003:** Effect of the cranberry extract on the live/dead cell ratio (i.e., the area occupied by living cells divided by the area occupied by dead cells) of the whole biofilm obtained by confocal laser scanning microscopy (CLSM). PBS: phosphate buffer saline; DMSO: 4% dimethyl sulfoxide solution.

	Treatment		Mean Difference (I–J)	Standard Error	Sig.^a^	95% Confidence Interval for Difference	
						Lower Bound	Upper Bound
**Antimicrobial effect**							
**30 s**	PBS	Cranberry	0.763	0.071	0.000	0.567	0.960
		DMSO	0.663	0.071	0.000	0.467	0.860
	Cranberry	DMSO	−0.100	0.071	0.550	−0.297	0.097
**60 s**	PBS	Cranberry	0.687	0.071	0.000	0.490	0.883
		DMSO	0.467	0.071	0.000	0.270	0.663
	Cranberry	DMSO	−0.220	0.071	0.027	−0.417	−0.023
**Anti-biofilm effect**							
**6 h**	PBS	Cranberry	0.35000	0.14575	0.160	−0.1292	0.8292
		DMSO	0.40000	0.14575	0.101	−0.0792	0.8792
	Cranberry	DMSO	0.05000	0.14575	1.000	−0.4292	0.5292

Based on estimated marginal means; ^a^
*p* value, adjustment for multiple comparisons (Bonferroni).

**Table 4 foods-09-00246-t004:** Anti-biofilm effects of the cranberry extract on the mean number of bacteria counts, incorporated during the 6 h of devolvement in the in vitro multi-species biofilm model (in colony forming units, CFUs mL^−1^, determined by quantitative real-time polymerase chain reaction (qPCR)). Data are expressed as mean and standard deviation (SD). PBS: phosphate buffer saline; DMSO: 4% dimethyl sulfoxide solution.

	Viable CFUs mL^−1^ [mean (SD)]			*p*-Value When Compared to Negative Control		% of Reduction of Viable CFUs mL^−1^ Respect to Negative Control	
	Negative Control (PBS)	Cranberry Extract	DMSO	Cranberry Extract	DMSO	Cranberry Extract	DMSO
***S. oralis***	1.2 × 10^5^ ± 2.5 × 10^4^	1.3 × 10^3^ ± 5.3 × 10^2^	5.5 × 10^2^ ± 2.6 × 10^2^	0.000	0.000	98.9	99.5
***A. naeslundii***	4.8 × 10^4^ ± 3.1 × 10^4^	7.8 × 10^4^ ± 7.6 × 10^4^	6.4 × 10^4^ ± 1.9 × 10^4^	0.608	1.000	-	-
***V. parvula***	2.3 × 10^4^ ± 1.5 × 10^4^	2.1 × 10^3^ ± 2.2 × 10^3^	2.0 × 10^4^ ± 7.3 × 10^3^	0.000	1.000	90.9	13.0
***A. actinomycetemcomitans***	7.5 × 10^5^ ± 2.8 × 10^5^	1.2 × 10^5^ ± 9.5 × 10^4^	3.8 × 10^5^ ± 1.4 × 10^5^	0.000	0.001	84.0	50.7
***P. gingivalis***	4.0 × 10^4^ ± 2.9 × 10^4^	1.1 × 10^3^ ± 1.1 × 10^3^	1.0 × 10^4^ ± 9.9 × 10^3^	0.000	0.0047	97.2	75.0
***F. nucleatum***	1.1 × 10^5^ ± 3.8 × 10^4^	2.7 × 10^4^ ± 2.0 × 10^4^	5.9 × 10^4^ ± 2.0 × 10^4^	0.000	0.005	75.4	46.4

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
