# Peer review of "New Evidences of Antibacterial Effects of Cranberry Against Periodontal Pathogens"

_foods, 2020, doi:10.3390/foods9020246_

Round 1

Reviewer 1 Report

The article “New evidences of antibacterial effects of cranberry against periodontal pathogens” by Sánchez and colleagues investigates the antibacterial and anti-biofilm activity of phenolic-rich cranberry extract in a validated in vitro biofilm model. The article is well-planned and written (despite minor errors listed at the end of this report). However, the results are not so encouraging if taking into account the (statistically significant) effect of 4% DMSO (correctly used as additional control), which is a major concern about the interpretation of the results. The authors must comment on this also in section 4 (Discussion) not just stating that the extract has an effect on the 6 bacterial species (e.g. lines 389-394, 421-425), as they correctly did in section 3 (Results). The authors should avoid the use DMSO (or at least diminish its %) in order to appreciate the true effect of the cranberry extract. At what degree the effect of the extract decreases if the % of DMSO is reduced to 3%, 2%, etc...is there a linear relationship or not?

Other minor points...

In section 2.1 the extraction procedure is reported but several details are missing, i.e. at what power were the samples sonicated? Please specify. “filtered through 0.22 μm” please reformulate properly.

In my opinion, sections 2.2 and 2.3 should be swapped to respect the chronological order: characterizations of the extract and then biological experiments.

Check compound names (i.e. 3-(3,4-Dihydroxyphenyl)-propionic acid (not 3.4), and 4-Hydroxyphenylacetic acid (without spaces) in Table 1).

Please convert rpm to g or rcf (i.e. line 185).

Please check superscripts, i.e. r2 must be corrected to r2 i.e in line 199; L-1 to L-1 in lines 326, 335, 351, etc.

p-coumaric (p italic) in line 253.

Check the bibliography format (i.e. line 495, journal title in italic).

Author Response

Please, see attached file.

Reviewer 2 Report

Dear Authors, 

Despite the fact that, the manuscript concerns a very interesting and important problem of the increasing resistance of microorganisms, as well as the alternative methods of biofilms eradication, it requires a significant improvement.

General remarks:

English must be improved, Add scheme with the analysis (it will be easier for the readers), There is no discussion on the compounds contained in cranberry.

Specific comments and suggestions are added in the PDF file. 

All the best, 

Author Response

Please, see attached file.

Reviewer 3 Report

Paper presents novel and interesting data. Methodology is well described which makes it repeatable. Results are clearly presented (Figures are fine).

Part regarding polyphenols could be improved. Cranberries contain both flavonoids and phenolic acids (please see phenol-explorer database), thus some general health effects of these polyphenols should be mentioned in the introduction (example: overall and cvd-related mortality (PMID: 28472215), diabetes (PMID: 29742713), cancer (PMID: 27943649))- worth to mention as all these outcomes are related to inflammation and to microbiota.

Author Response

Please, see attached file.

Round 2

Reviewer 1 Report

The revised version of the manuscript is significantly improved and, although my concern about the influence of the solvent (DMSO) on the results still persists (and it should be investigated in deeper detail in future works...), in my opinion it now deserves publication in Foods.

Reviewer 2 Report

Dear Authors, 

thank you for the new version of the manusript. In my oppinion it was improved. 

All the best,